# Knowledge and Attitudes of Cannabidiol in Croatia among Students, Physicians, and Pharmacists

**DOI:** 10.3390/pharmacy12010002

**Published:** 2023-12-23

**Authors:** Ana Batinic, Ana Curkovic, Josipa Bukic, Irena Žuntar, Sendi Kuret, Bianka Mimica, Nina Kalajzic, Goran Dujic, Ljubica Glavaš-Obrovac, Ana Soldo, Andrijana Včeva, Zeljko Dujic, Davorka Sutlovic

**Affiliations:** 1Pharmacy of Split-Dalmatia County, 21000 Split, Croatia; 2Department of Health Studies, University of Split, 21000 Split, Croatia; acurkovic@ozs.unist.hr (A.C.); sendikuret@gmail.com (S.K.); nkalajzic@ozs.unist.hr (N.K.); dsutlovic@ozs.unist.hr (D.S.); 3Department of Pharmacy, University of Split School of Medicine, 21000 Split, Croatia; josipa.bukic@mefst.hr; 4Faculty of Pharmacy and Biochemistry, University of Zagreb, 10000 Zagreb, Croatia; irena.zuntar@pharma.unizg.hr; 5School of Medicine, University of Split School of Medicine, 21000 Split, Croatia; 6Clinical Department of Diagnostic and Interventional Radiology, University Hospital of Split, 21000 Split, Croatia; goran.dujic@gmail.com; 7Department of Medicinal Chemistry, Biochemistry and Clinical Chemistry, Faculty of Medicine, Josip Juraj Strossmayer University of Osijek, 31000 Osijek, Croatia; lgobrovac@mefos.hr; 8Croatian Chamber for Pharmacists, 10000 Zagreb, Croatia; ana.soldo@hljk.hr; 9Department of Otorhinolaryngology and Maxillofacial Surgery, Medical Faculty, University of Osijek, J. Huttlera 4, 31000 Osijek, Croatia; avceva@mefos.hr; 10Department of Otorhinolaryngology and Head and Neck Surgery, Clinical Hospital Centre Osijek, J. Huttlera 4, 31000 Osijek, Croatia; 11Department of Integrative Physiology, School of Medicine, University of Split, 21000 Split, Croatia; zeljko.dujic@mefst.hr; 12Department of Toxicology and Pharmacogenetics, School of Medicine, University of Split, 21000 Split, Croatia

**Keywords:** cannabidiol, knowledge, attitudes, pharmacists, physicians, students, Croatia

## Abstract

Due to cannabidiol’s health benefits and absence of serious side effects, its use is constantly growing. This is a survey-based cross-sectional study that was conducted to determine Croatian pharmacists’, physicians’, and students’ knowledge and attitudes about cannabidiol (CBD). Two questionnaires were created, one for students and the other for physicians and pharmacists. Our participants (in total 874: 473 students and 401 physicians and pharmacists) generally had positive attitudes towards CBD therapy as approximately 60% of them believe that CBD treatment is generally efficacious. Participants had positive attitudes toward the therapeutic value of CBD, especially pharmacists and pharmacy students (63.8% and 72.2%, respectively). Pharmacists were significantly more convinced that CBD could reduce the use of opioids prescribed for chronic pain (*p* < 0.05). Only 17.5% of students had read scientific papers about CBD, compared to a significantly higher percentage of physicians and pharmacists (43.0% and 47.8%, respectively) (*p* < 0.05). This study revealed a gap in knowledge regarding CBD, since 89.3% of pharmacists and physicians, as well as 84.8% of students, believe they need more education about CBD. We conclude that it is important to improve the educational curricula so that medical professionals can recommend CBD use to their patients when needed.

## 1. Introduction

The hemp plant (lat. *Cannabis sativa* L.) is a widely used plant, and its subspecies Indian hemp (lat. *Cannabis sativa* L. *subsp. indica*), i.e., marijuana, is considered one of the most commonly consumed recreational drugs worldwide, especially among young adults [1,2]. There are more than 750 identified cannabis chemicals, including more than 100 cannabinoids. Cannabidiol (CBD) and delta-tetrahydrocannabidiol (THC) are the two most important and widely studied components [3,4]. Cannabis is associated with recreational drugs due to THC, which is known as the primary psychoactive component of the plant [5]. However, CBD has no psychotropic effects and has a confirmed safety profile [6,7]. Due to its numerous health benefits and lack of significant negative side effects, CBD use and product marketing are constantly increasing [8,9]. National regulations for the use of CBD vary around the world. The use of CBD as a dietary supplement is allowed in many countries as long as the THC content is below 0.3% in the United States and 0.2% in Europe [10]. 

Currently, the Food and Drug Administration (FDA) has approved only one purified, prescription CBD medicine (Epidiolex^®^, 100 mg/mL, oral solution). This drug has been designated as an “orphan drug” (a medication used to treat rare disorders). Epidiolex is indicated as adjunctive therapy for seizures associated with Lennox-Gastaut syndrome (LGS) or Dravet syndrome (DS) in combination with clobazam in patients ≥ 2 years old and as adjunctive therapy for seizures associated with tuberous sclerosis complex (TSC) also in patients ≥ 2 years old [11,12,13]. Marinol^®^ (dronabinol), Syndros^®^ (dronabinol), and Cesamet^®^ (nabilone) are three synthetic cannabis-related pharmacological products, also approved by the FDA. Dronabinol is a synthetic delta-9- tetrahydrocannabinol (THC), which is considered the psychoactive intoxicating component of cannabis (i.e., the component responsible for the “high” people can experience when using cannabis). The use of dronabinol is indicated for nausea and vomiting associated with malignancies and for the treatment of anorexia associated with weight loss in patients with acquired immunodeficiency syndrome (AIDS) [11,14]. Nabilone (a synthetic with a THC-like chemical structure) is indicated for the treatment of nausea and vomiting associated with cancer chemotherapy in patients who have not responded adequately to conventional antiemetic treatments. These medications are available in the United States only with a prescription from a licensed healthcare provider [11]. The European Medicines Agency (EMA), has approved the use of Epidyolex^®^ (cannabidiol) for the same indications accepted by the FDA [15]. CBD is marketed as Epidyolex in the European Union, but it is officially known as Epidiolex in the USA. In addition, the EMA has also approved Sativex^®^, an oromucosal spray (solution), containing two extracts of *Cannabis sativa* L., folium cum flore (cannabis leaf and flower), which contain almost the same amount of THC and CBD [16]. Sativex is indicated as a treatment to improve symptoms in adult patients with moderate to severe spasticity due to multiple sclerosis (MS) who have not responded adequately to other anti-spasticity medications and who show clinically significant improvement in symptoms associated with spasticity during an initial trial of therapy. The FDA has not yet approved Sativex in the United States.

An increasing body of evidence-based information available, including multiple CBD human research, now supports the long-standing use of cannabis and CBD products to treat a variety of medical conditions: symptoms of chronic pain, inflammation, cardiovascular disease, mental health issues, spasticity associated with multiple sclerosis and malignancies without serious side effects [17,18]. With the development of novel CBD formulations, smaller doses may lead to increased absorption and, consequently, greater health benefits [19,20,21,22,23].

Nowadays, people are becoming more aware and interested in the natural medicinal aspects of CBD, as it is becoming more widely available in cosmetics and dietary supplements. In Croatia, the Agency for Medicinal Products and Medical Devices (HALMED) has approved the FDA and EMA-approved cannabinoid-based medication Epidyolex, and all relevant details about the medicine, including interactions with other medicines, are available [24]. The use of unapproved cannabis and cannabis-derived products may have unpredictable and unintended consequences, including serious safety risks, considering that there are CBD products of questionable quality and with inconsistent labelling on the market [11,25]. According to recent studies, young adults have a positive perception of CBD despite having limited knowledge of its evidence base or regulation [26]. Several studies have shown that the pharmacological knowledge of pharmacists, physicians, students, patients, and recreational users in other countries is insufficient regarding cannabis and cannabinoid-derived drugs [12,26,27,28,29,30,31,32]. This study aimed to analyze the attitudes and knowledge of physicians, pharmacists, and students in Croatia about the therapeutic use of cannabis and cannabinoid-derived medicines.

## 2. Materials and Methods

A cross-sectional survey study was conducted from 30 June to 30 July 2023. Two questionnaires were developed for this study, one to assess the knowledge and attitudes of physicians and pharmacists about the use of CBD for medical purposes and another to assess the knowledge and attitudes of students. The sample of students included medical, pharmacy, and health science students. Both questionnaires were developed by the researcher and were based on a literature review of this particular topic [10,12,28,32,33,34,35,36,37,38,39,40,41,42,43,44,45,46].

### 2.1. Surveys Design

The questionnaire for the students consisted of 20 questions and the physicians’ and pharmacists’ questionnaire consisted of 31 questions. The questions were divided into 5 categories: general questions, self-assessment knowledge questions, researcher-identified knowledge questions, CBD experience questions, and attitude assessment questions about CBD use (Table 1). Attitudes and knowledge regarding CBD were assessed using a 5-point Likert scale (from strongly disagree to strongly agree), yes/no questions, and categorical questions (with one or more choices).

Surveys were created and distributed using Google Forms online survey administration software offered by Google. The open survey link was sent to physicians and pharmacists across Croatia and students at the Universities of Split (medical, pharmacy, and health students), Zagreb (pharmacy students), and Osijek (medical students). The sample size was determined using the SurveyMonkey sample size calculator [47]. The confidence level was 95% with a margin of error of 5%.

With a target population of 2000 students and 20,000 physicians and pharmacists, the required sample was 323 for students and 377 for physicians and pharmacists. The final sample consisted of 874 participants, of whom 473 were students and 401 were physicians and pharmacists. 

The survey was completely anonymous and voluntary, and it was approved by the Ethical Committee of the University Department of Health studies at the University of Split on 26 June 2023 (Class: 029-03/23-08/01; Registration number: 2181-228-103/1-47).

### 2.2. Statistical Analysis

Data analysis utilized descriptive statistics to describe responses to survey items. The differences between the groups of study parameters were measured using the Chi-square and Mann-Whitney U tests. Chi-square tests were utilized for comparisons of common perceptions about knowledge and education about CBD between physicians, pharmacists, and surveyed students. Differences and relationships were considered to be statistically significant at *p*-value < 0.05. Statistical analysis was performed using Statistical Package Software for Social Science, version 26 (SPSS Inc., Chicago, IL, USA).

## 3. Results

### 3.1. Demographic Data

General statistics of study participants who completed the online survey were presented in Table 2 and Table 3. Among demographic data, tables showed participants’ knowledge self-assessment and researcher-assesses knowledge. Our study consisted of a total of 874 participants; students (*N* = 473), physicians’ and pharmacists’ [*N* = 401: 100 physicians (52 specialists and 48 general practitioners) and 301 pharmacists (16 specialized pharmacists and 285 pharmacists without specialization)]. The majority of all respondents were female, as in other similar research [12,32,36]. The percentage of female students was 78.65% and the percentage of males was 21.35%. The Croatian Bureau of Statistics reports that 151,827 students were enrolled for the academic year 2022/2023 57.9% of them were female students, and 42.1% were males [48]. The study included responses from 100 physicians from both genders: females (70.0%) and males (30.0%). Among pharmacists’ the proportion of female participants was 85.0% and for males was 15.0%. A representative sample of participants took part in the survey, the Croatian Medical Chamber has 16,089 members (63% female), and the Croatian Chamber of Pharmacists has 4325 members (88.6% female) [49,50].

Participants were not paid for their participation as it was voluntary. Croatian medical studies last six years, while pharmacy and public health studies have five-year programs (with the exception of the 3-year basic public health studies). Croatian students from all years of study enrolled in the academic year 2022/2023 were included in this survey.

Respondents include physicians and pharmacists with a wide range of professional experiences, from one to more than forty years.

### 3.2. Results of Respondent’s Knowledge about CBD

In our study, we did not find any differences between groups of participants where more than 70% (76.3% and 78.8%) of respondents believe they have general knowledge about CBD (Figure 1 and Table 2). Interestingly, more than 70% (73.2% and 77.6%) of them answered that they did not have a formal education about it. 

The majority of participants, with the exception of pharmacists, were generally neutral regarding whether CBD is harmful to health. A significant difference in perceptions of CBD’s hazards was not observed between all student groups (*p* = 0.059) while a statistical difference was observed between students and a group of physicians and pharmacists (as well as between a group of physicians and pharmacists) (*p* < 0.05). For participants agreeing with level 1 strongly disagree, there was a statistically significant difference between all respondents’ responses (*p* < 0.05) (Figure 2A and Table 3). 

A significant difference (*p* < 0.05) was found regarding attitudes toward the efficacy of CBD therapy between all groups of students (Figure 2B) and between pharmacists and physicians. A statistically significant difference was also observed between the responses of all respondents for participants’ agreement level 2-disagree and 5-strongly agree (*p* < 0.05). (Figure 2B). In general, participants had high and very high attitudes toward the therapeutic value of CBD, especially pharmacists and pharmacy students (63.8% and 72.2%, respectively).

### 3.3. Results of the Questionnaire Presented Only for Physicians’ and Pharmacists’

In Figure 3, Figure 4, Figure 5, Figure 6 and Figure 7 and Table 4 results of the questionnaire presented only for physicians and pharmacists were shown. Figure 3 and Figure 4 showed no significant difference between the group of physicians and pharmacists regarding the FDA-approved indications for CBD as well as their knowledge of CBD side effects.

Additionally, as shown in Figure 5. there was not a significant difference between the groups of physicians and pharmacists regarding knowledge about CBD and drug interactions. Also, physicians and pharmacists are aware of the need for caution while utilizing CBD for specific medical conditions except for reduced body weight (only 11% of physicians and 20% of pharmacists) as it was shown in Figure 6.

Finally, there wasn’t a significant difference in the attitudes between the two groups of participants, physicians, and pharmacists, regarding support of the use of CBD for different medical conditions as it was presented in Figure 7. Most pharmacists and physicians support the use of CBD for malignant conditions and in palliative patients. Furthermore, pharmacists are significantly more convinced that CBD could reduce the use of opioids prescribed for chronic pain, as it was shown in Table 4. Both test groups generally agree that they do not have enough knowledge about the use of CBD for medical purposes and therefore can’t recommend it to patients. Pharmacists and physicians support health insurance coverage for CBD use.

## 4. Discussion

### 4.1. Questions Presented in Both Questionnaires

To the best of our knowledge, this is the first study that explored perceptions and knowledge regarding the therapeutic use of CBD among students in Split, Zagreb, and Osijek, as well as among pharmacists and physicians in Croatia. 

This study revealed a gap in knowledge regarding CBD, among both groups since 89.3% of pharmacists and physicians, as well as 84.8% of students, believe they need more education about CBD. As in previous studies [2,12,28,31], most of our respondents also believe that curricula should include lectures on the use of CBD for medicinal purposes. In addition, we also find that 63.6% of students and 54.6% of pharmacists and physicians agreed that taking CBD as therapy is beneficial; this finding is consistent with published research in which participants revealed generally positive attitudes toward medical cannabis therapy [8,27]. Participants in the Schilling et al.’s study [27] revealed a positive attitude toward CBD products as a therapeutic alternative, as they reported positive outcomes and expressed an interest in learning more about CBD from their physicians. Approximately 40% of all our participants believe that CBD use has positive effects on physical and mental health, while about 60% of them believe that CBD treatment is generally efficacious.

However, Goodman et al. [8] observed that little is known about the potential negative effects of CBD. According to our study, only 31.5% of students and 26.2% of physicians and pharmacists considered they were aware of the risks associated with CBD use.

Almost a quarter of all respondents have no knowledge of CBD. This is unexpected among healthcare professionals, considering how widespread CBD products are on the market today. This information is also unexpected from an academic perspective, as nowadays over 500 research on potential indications of CBD have been reported on ClinicalTrials.gov, a well-known website and online database of clinical research studies and information about their results that provide information to the public, researchers and health care professionals (https://clinicaltrials.gov/, accessed on 2 December 2023). The results of our study are consistent with the results of a nationwide survey on CBD use and attitudes in France, where 30% of participants had never heard of CBD [10]. Regarding a question about reading scientific literature on CBD, there was a significant difference between groups as only 17.5% of students had read scientific papers about CBD, compared to a significantly higher percentage of physicians and pharmacists (43% and 47.8%, respectively). 

Due to students’ significantly greater CBD use than physicians and pharmacists, the results confirm our expectations that CBD consumption is associated with students, who are a younger age group than professionals, as was shown in previous studies [10,51]. Our results show that among all students, pharmacists (84.8%) have the most knowledge about CBD, followed by health (73.6%) and medical students (70.7%). This result is in contrast to the same conducted in Austria where medical students had the most knowledge about CBD [28].

### 4.2. Specific Knowledge of CBD among Physicians and Pharmacists

Based on the study’s results, physicians and pharmacists frequently link the FDA-approved indications for CBD with the ones for dronabinol and nabilone. However, the indication for epileptic seizures in LGS (Lennox-Gastaut syndrome) and Dravet syndrome was more often recognized by pharmacists (36%) than by physicians (23%), with participants mostly unaware that tuberous sclerosis is also among indications [24].

Physicians and pharmacists more often indicated pain associated with malignant diseases, chronic neuropathic pain, and chemotherapy-related nausea as a possible FDA-approved indication, although FDA-approved indications are only seizures associated with Lennox-Gastaut syndrome (LGS) or Dravet syndrome (DS) and for seizures associated with tuberous sclerosis complex.

Somnolence, reduced appetite, diarrhea, and vomiting are the most commonly reported side effects of CBD. Participants’ knowledge was generally good, with the exception of tachycardia, which was selected by more than one-third of physicians and pharmacists. Previous studies have shown that CBD lowers heart rate, diastolic pressure, and MAP (mean arterial pressure) without causing tachycardia [19,52]. Participants mainly recognize carbamazepine, valproate, rifampicin, clobazam, and everolimus as drugs that have the greatest interactions with CBD. Although there was no significant difference in knowledge, pharmacists were more familiar with interactions. Medical conditions that require caution when taking CBD (somnolence and sedation, suicidal behaviour, hepatocellular damage) were generally well-known to respondents. They are mostly unaware that caution is required even with reduced body weight (only 20% of pharmacists and 11% of physicians are aware) and that heart arrhythmia does not require caution when dosing CBD (43% of pharmacists and 48% of physicians).

In contrast to physicians, pharmacists are significantly more likely to believe that recommending/prescribing CBD could reduce opioid use for chronic pain, as some research suggests [53]. In their study, McNabb et al. [53] proved that the consumption of pharmaceutical medications and other substances by veterans could potentially be reduced due to medicinal cannabis. Physicians and pharmacists agree that they do not have enough knowledge about the use of CBD for medical purposes and, therefore, cannot recommend it to their patients (94% and 88.7%, respectively).

Knowledge about CBD was found to be insufficient among medical students and healthcare professionals in the prior studies [12,28,31,32,54]. Participants agree that health insurance should cover the cost of CBD when a physician prescribes it as a therapy. Epidyolex is now available only with a restricted prescription and is entirely paid for by the patient in Croatia (more than 12 hundred euros per bottle of 100 mL) [24].

### 4.3. Study Limitations

This cross-sectional study has certain limitations. Despite the representative sample of participants, we were limited to a small percentage of physicians compared to the total number of physicians in Croatia, in contrast to the substantial number of participants- pharmacists. With the large final sample size, we believe that the effects were partially reduced. Another limitation was that students from other biomedical faculties in Croatia were not included in the survey.

## 5. Conclusions

The results of our survey indicate that current and future healthcare professionals involved in the process of patients’ medication, medical, pharmacy, and health students, as well as physicians and pharmacists, believe they need additional education on the proper and safe use of CBD. Therefore, it is indispensable to improve the educational curricula so that medical professionals have more knowledge and can recommend CBD use to their patients when needed. Nevertheless, physicians and pharmacists have shown that although they have close to enough knowledge about the indications, side effects, and interactions of CBD, they hardly prescribe and/or recommend it.

We assume that the reason for this, in addition to the uncertainty in knowledge, is the high price of the product. Therefore, it is understandable that physicians and pharmacists generally support that health insurance should cover the cost of the medicine. Further research is required to gain a more comprehensive understanding of the specific challenges and factors influencing knowledge gaps in these areas.

## Figures and Tables

**Figure 1 pharmacy-12-00002-f001:**
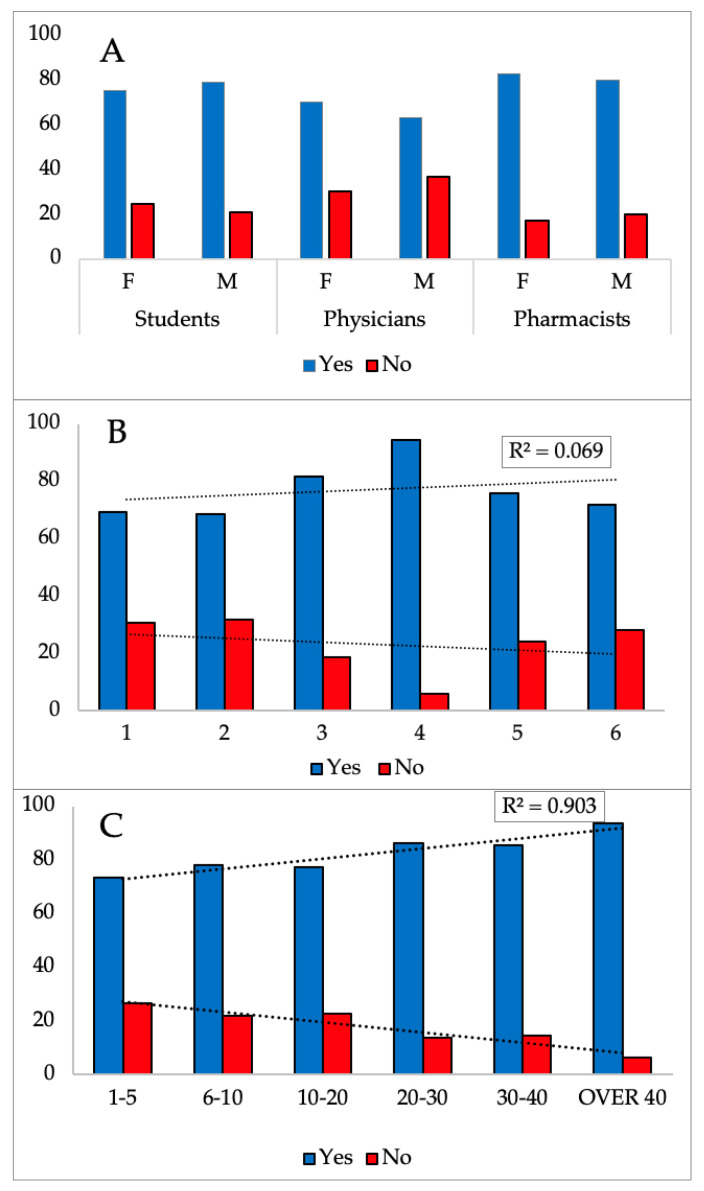
Percentage of participants’ knowledge about CBD. On the y-axis, the percentage of respondents’ responses is displayed. The x-axis is shown as follows: (**A**) shows the distribution of respondents by gender (F—female and M—male) for the student, doctor, and pharmacist groups; (**B**) shows the respondents’ years of study for the student group; and (**C**) shows the respondents’ number of years of working experience for the doctor and pharmacist group. The answer trend Yes is positively correlated: in the examined group of students from 1 to 4 years of study (**B**) and in the group of doctors and pharmacists depending on the years of work experience.

**Figure 2 pharmacy-12-00002-f002:**
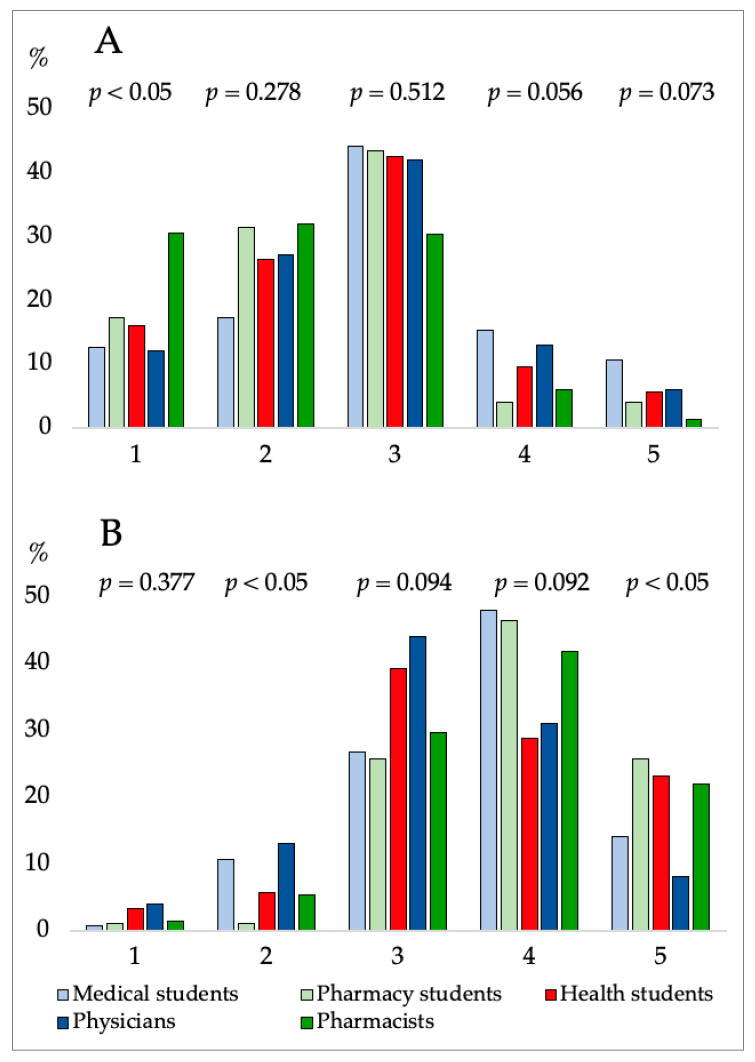
Percentage of participants’ attitudes regarding questions: (**A**)—shows the answers to the questions “CBD is bad for health”, (**B**)—shows the answers to the questions “CBD treatment is efficacious” Difference between groups: χ^2^ Test. On the y-axis, the percentage of respondents’ responses is displayed. On the y-axis is displayed participants agreement level: 1—strongly disagree; 2—disagree; 3—neutral; 4—agree; and 5—strongly agree.

**Figure 3 pharmacy-12-00002-f003:**
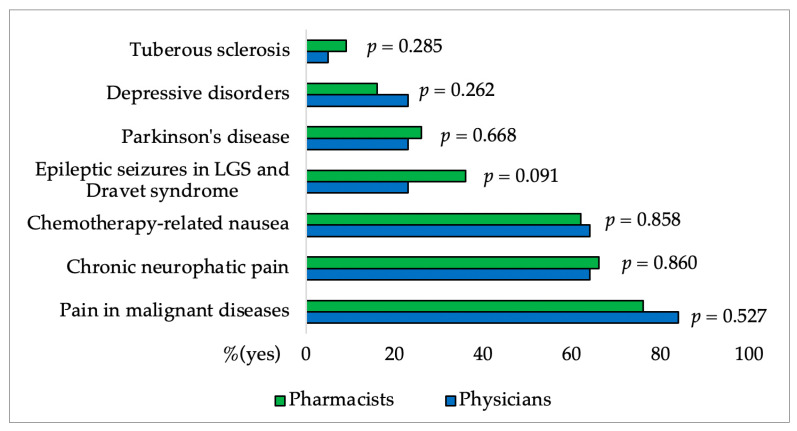
Percentages of pharmacists’ and physicians’ affirmative responses regarding the FDA-approved indications for CBD (LGS: Lennox-Gastaut syndrome). The difference between groups for each indication was made using the χ^2^ Test.

**Figure 4 pharmacy-12-00002-f004:**
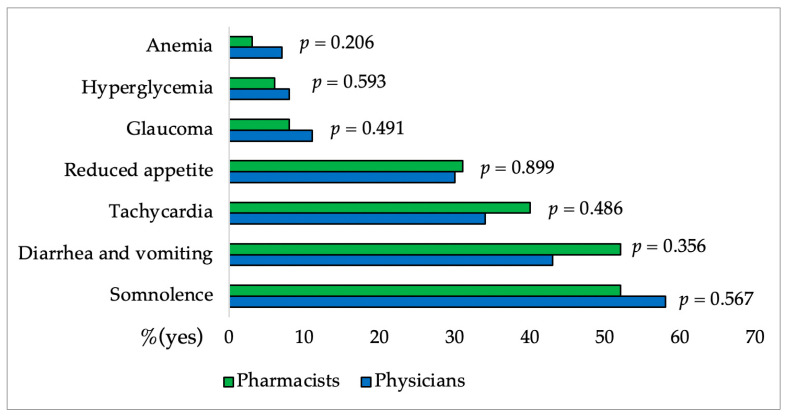
Percentages of pharmacists and physicians’ affirmative responses regarding the CBD side effects. The difference between groups for each side effect was made using the χ^2^ Test.

**Figure 5 pharmacy-12-00002-f005:**
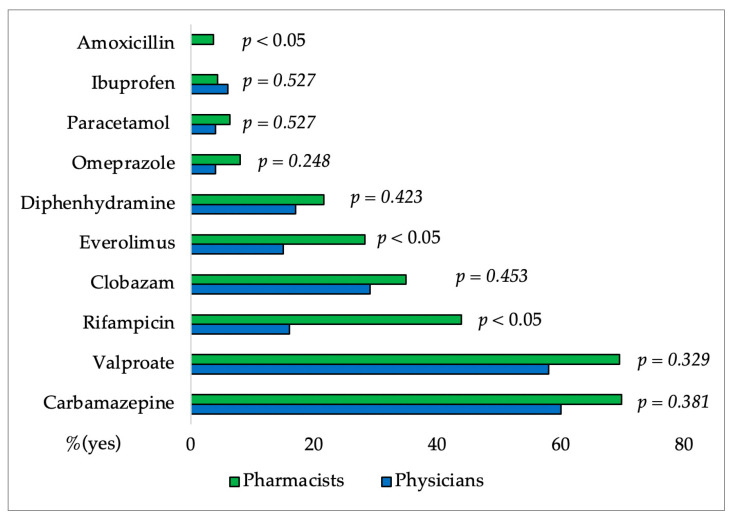
Percentages of pharmacists and physicians’ affirmative responses regarding CBD and drug interactions. The difference between groups for each drug interaction was made using the χ^2^ Test.

**Figure 6 pharmacy-12-00002-f006:**
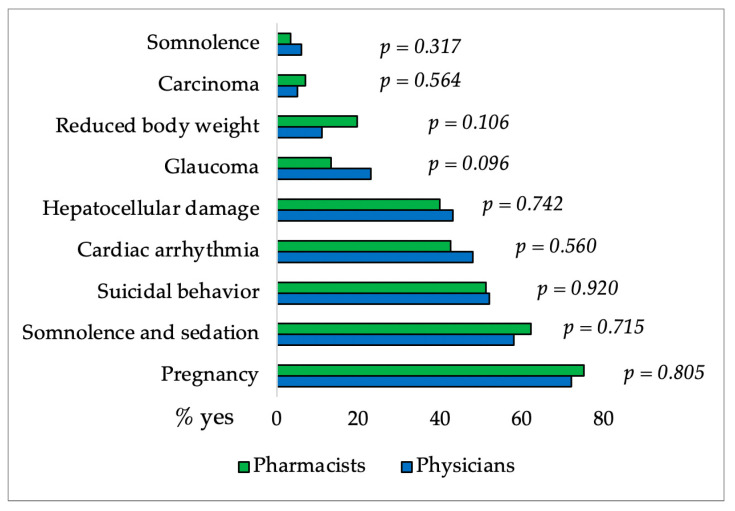
Percentages of pharmacists’ and physicians’ affirmative responses regarding different medical conditions when ingestion of CBD requires caution. The difference between groups for each medical condition was made using the χ^2^ Test.

**Figure 7 pharmacy-12-00002-f007:**
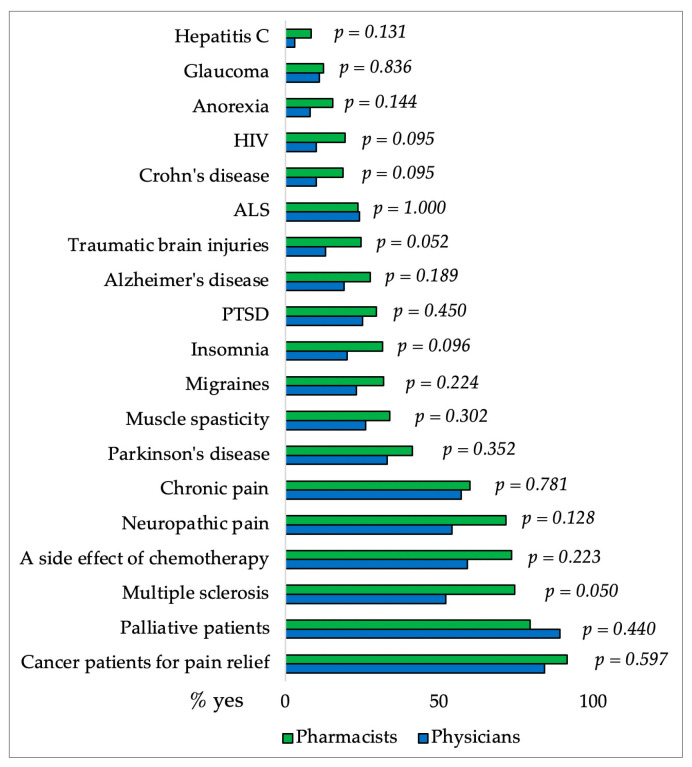
Percentages of pharmacists and physicians who support the use of CBD for different medical conditions. The difference between groups for each medical condition was made using the χ^2^ Test.

**Table 1 pharmacy-12-00002-t001:** Surveys design: categories and questions for respondents.

Question Category	Physicians’ and Pharmacist’ Questionnaire	Students’ Questionnaire
General	Gender, profession, specialization, years of work in practice, county of residence	Gender, study program, year of study
Questions represented in both questionnaires		
Knowledge self-assessment	Do you have knowledge about CBD? Through my formal education, I had an education about CBD. I think that I need more education about CBD. I am aware of CBD use risks. I am aware of CBD use benefits.	
Researcher-assessed knowledge	CBD is bad for health. CBD treatment is efficacious. CBD has positive effects on physical health. CBD has positive effects on mental health. CBD helps patients with chronically debilitating conditions. CBD is physically addictive. CBD is psychologically addictive. Using CBD can lead to addiction to other opioids and drugs. CBD causes a feeling of euphoria. Have you ever read a scientific paper about CBD?	
CBD experience	Have you ever consumed CBD?	
Attitudes about CBD use	The educational curricula of Physicians, health professionals, and pharmacists should include subjects on the use of CBD for medical purposes.	
Questions presented in physicians’ and pharmacists’ questionnaire only		
Knowledge self-assessment	I believe that I have enough knowledge about the use of CBD for medical purposes and that I can recommend it to patients.	
Researcher-assessed knowledge	FDA has approved CBD drugs for nausea associated with chemotherapy, chronic neuropathic pain, EPI attacks in Lennox-Gastaut and Dravet syndrome, depressive disorders, Parkinson’s disease, tuberous sclerosis, and pain in malignant diseases. ** Side effects of CBD are anemia, tachycardia, diarrhea and vomiting, glaucoma, decreased appetite, hyperglycemia, and somnolence. ** Medications that have moderate or severe interactions with CBD include: paracetamol, valproat, omeprazole, karbamazepin, ibuprofen, rifampicin, amoksicilin, everolimus, klobazam, fehidramin. ** Conditions that require caution when using CBD are cardiac arrhythmia, hepatocellular damage, glaucoma, somnolence, cancer, reduced body weight, pregnancy, suicidal behaviour, somnolence, and sedation. **	
CBD experience	Have you ever recommended/prescribed CBD to your patients?	
Attitudes about CBD use	I support the use of CBD in palliative patients, cancer pain relief, side effects of chemotherapy, multiple sclerosis, neuropathic pain, chronic pain, PTSD, insomnia, Crohn’s disease, glaucoma, hepatitis C, muscle spasticity, HIV, traumatic brain injury, ALS, Alzheimer’s disease, anorexia, Parkinson’s disease, migraine. ** I believe that recommending/prescribing CBD could reduce the use of opioids in chronic pain. I believe that health insurance should cover the cost of CBD if a doctor prescribes it as therapy.	

** Check all that apply.

**Table 2 pharmacy-12-00002-t002:** Study sample general characteristics and knowledge self-assessment of Croatian students (*N* = 473), physicians (*N* = 100), and pharmacists (*N* = 301).

Variable		Students *N* (%)	Physicians’ and Pharmacists’ *N* (%)	Difference between Groups χ^2^ Test
Gender	F	372 (78.6)	326 (81.3)	*p* = 0.330
	M	101 (21.4)	75 (18.7)	
Study program	Medical	150 (31.7)	-	
	Pharmacy	198 (41.9)	-	
	Health	125 (26.4)	-	
Year of study program Medical 1–6; Pharmacy 1–5	1	111 (23.5)	-	
	2	92 (19.5)	-	
	3	92 (19.5)	-	
	4	71 (15.0)	-	
	5	50 (10.6)	-	
	6	57 (12.1)	-	
Years of work in practice physicians’/pharmacists’ (*N* = 401)	1–5	-	135 (33.7)	
	6–10	-	50 (12.5)	
	11–20	-	101 (25.2)	
	21–30	-	65 (16.2)	
	31–40	-	34 (8.5)	
	>40	-	16 (4.0)	
Do you have knowledge about CBD?	Yes	361 (76.3)	316 (78.8)	*p* = 0.382
	No	112 (23.7)	85 (21.2)	
Through my formal education, I had an education about CBD.	Yes	127 (26.8)	90 (22.4)	*p* = 0.136
	No	346 (73.2)	311 (77.6)	
Have you ever read a scientific paper about CBD?	Yes	83 (17.5)	187 (46.6)	*p* < 0.05
	No	390 (82.5)	214 (53.4)	
Have you ever consumed CBD?	Yes	120 (25.4)	65 (16.2)	*p* < 0.05
	No	353 (74.6)	336 (83.8)	
The educational curricula of Physicians, health professionals, and pharmacists should include subjects on The use of CBD for medical purposes.	1	11 (2.3)	14 (3.5)	*p* < 0.05
	2	15 (3.2)	9 (2.2)	
	3	67 (14.2)	39 (9.7)	
	4	143 (30.2)	85 (21.2)	
	5	237 (50.1)	254 (63.3)	
I think that I need more education about CBD.	1	17 (3.6)	10 (2.5)	*p* < 0.05
	2	11 (2.3)	15 (3.7)	
	3	44 (9.3)	18 (4.5)	
	4	108 (22.8)	61 (15.2)	
	5	293 (61.9)	297 (74.1)	
I am aware of CBD use risks.	1	88 (18.6)	61 (15.2)	*p* < 0.05
	2	96 (20.3)	78 (19.5)	
	3	140 (29.6)	157 (39.2)	
	4	91 (19.2)	70 (17.5)	
	5	58 (12.3)	35 (8.7)	
I am aware of CBD use benefits.	1	50 (10.6)	33 (8.2)	*p* < 0.05
	2	75 (15.9)	36 (9.0)	
	3	144 (30.4)	162 (40.4)	
	4	155 (32.8)	136 (33.9)	
	5	49 (10.4)	34 (8.5)	

Participants agreement level: 1—strongly disagree; 2—disagree; 3—neutral; 4—agree; and 5—strongly agree.

**Table 3 pharmacy-12-00002-t003:** Researcher assessment of knowledge and differences between analyzed groups: Croatian students (*N* = 473), physicians (*N* = 100), and pharmacists (*N* = 301).

Variable	* Participants Agreement Level	Students *N* (%)	Physicians’ and Pharmacists’ *N* (%)	Difference between Groups χ^2^ Test
CBD is bad for health.	1	73 (15.4)	104 (25.9)	*p* < 0.05
	2	121 (25.6)	123 (30.7)	
	3	205 (43.3)	133 (33.2)	
	4	43 (9.1)	31 (7.7)	
	5	31 (6.6)	10 (2.5)	
CBD treatment is efficacious.	1	7 (1.5)	8 (2.0)	*p* = 0.408
	2	25 (5.3)	29 (7.2)	
	3	140 (29.6)	133 (33.2)	
	4	200 (42.3)	157 (39.2)	
	5	101 (21.4)	74 (18.5)	
CBD has positive effects on physical health.	1	21 (4.4)	11 (2.7)	*p* = 0.158
	2	37 (7.8)	29 (7.2)	
	3	225 (47.6)	168 (41.9)	
	4	140 (29.6)	145 (36.2)	
	5	50 (10.6)	48 (12.0)	
CBD has positive effects on mental health.	1	41 (8.7)	21 (5.2)	*p* = 0.092
	2	75 (15.9)	49 (12.2)	
	3	181 (38.3)	164 (40.9)	
	4	126 (26.6)	127 (31.7)	
	5	50 (10.6)	40 (10.0)	
CBD helps patients with chronically debilitating conditions.	1	5 (1.1)	5 (1.2)	*p* = 0.546
	2	18 (3.8)	12 (3.0)	
	3	120 (25.4)	112 (27.9)	
	4	194 (41.0)	175 (43.6)	
	5	136 (28.8)	97 (24.2)	
CBD is physically addictive.	1	74 (15.6)	92 (22.9)	*p* < 0.05
	2	98 (20.7)	89 (22.2)	
	3	168 (35.5)	143 (35.7)	
	4	88 (18.6)	46 (11.5)	
	5	45 (9.5)	31 (7.7)	
CBD is psychologically addictive.	1	40 (8.5)	71 (17.7)	*p* < 0.05
	2	58 (12.3)	70 (17.5)	
	3	164 (34.7)	136 (33.9)	
	4	125 (26.4)	77 (19.2)	
	5	86 (18.2)	47 (11.7)	
Using CBD can lead to addiction to other opioids and drugs.	1	87 (18.4)	126 (31.4)	*p* < 0.05
	2	88 (18.6)	91 (22.7)	
	3	155 (32.8)	115 (28.7)	
	4	87 (18.4)	38 (9.5)	
	5	56 (11.8)	31 (7.7)	
CBD causes a feeling of euphoria.	1	81 (17.1)	118 (29.4)	*p* < 0.05
	2	86 (18.2)	100 (24.9)	
	3	177 (37.4)	126 (31.4)	
	4	77 (16.3)	38 (9.5)	
	5	52 (11.0)	19 (4.7)	

* Participants agreement level: 1—strongly disagree; 2—disagree; 3—neutral; 4—agree; and 5—strongly agree.

**Table 4 pharmacy-12-00002-t004:** Differences between physicians’ and pharmacists’ attitudes and knowledge about prescribing/recommending the use of CBD. The difference between groups was made using the χ^2^ Test.

Variable		Physicians *n* (%)	Pharmacists *n* (%)	Differences between Groups (χ^2^ Test)
Question 25 I believe that recommending/prescribing CBD could reduce the use of opioids in chronic pain.	Yes	6 (6)	251 (83.4)	*p* < 0.05
	No	94 (94)	50 (16.6)	
Question 26 I believe that I have enough knowledge about the use of CBD for medical purposes and that I can recommend it to patients.	Yes	6 (6)	34 (11.3)	*p* = 0.126
	No	94 (94)	267 (88.7)	
Question 27 Have you ever recommended/prescribed the use of CBD to patients in your practice so far?	Yes, just once	5 (5)	25 (8.3)	*p* = 0.108
	Yes, more than once	1 (1)	19 (6.3)	
	Yes, often to patients with specific diagnoses	2 (2)	7 (2.3)	
	No	92 (92)	250 (83.1)	
Question 28 I believe that healthcare insurance should cover the cost of CBD if a doctor prescribes it as therapy.	Yes	83 (83)	256 (85.0)	*p* = 0.623
	No	17 (17)	45 (15.0)	

## Data Availability

Data are contained within the article and Appendix A.

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
