# Peer review of "Knowledge and Attitudes of Cannabidiol in Croatia among Students, Physicians, and Pharmacists"

_pharmacy, 2023, doi:10.3390/pharmacy12010002_

Round 1
Reviewer 1 Report
Comments and Suggestions for Authors
The manuscript “Knowledge and attitudes of cannabidiol in Croatia among students, physicians, and pharmacists” addresses a timely and important question, is well-powered, and generally cautiously interpreted. The biggest concern is with the data-analysis (Point B). Once revised, the updated discussion should take care to both highlight the general pattern in the data (as was done) but also to identify some findings where there were statistically significant group differences.
General concerns
A. The description of the methods is generally ok/good. Please read:
Eysenbach G: Improving the quality of Web surveys: the Checklist for Reporting Results of Internet E-Surveys (CHERRIES). J Med Internet Res 2004, 6:e34.
Incorporate its suggestions, and cite this paper (or another if a more recent equivalent is available).
For transparency and future replication/extension efforts, it would be quite useful if the full original surveys (in Croatian?) could also be included in the Supplemental Materials.
B. The statistics are either unusual in places or were not fully completed. Showing the % of each group in Fig 2 with each of 5 responses, is not very informative. Consider running parametric analyses (exploratory t-tests on means) or non-parametric (chi-sq on % high/very high).
For Fig 3, consider revisiting this with the statistician on the team. Mann-Whitney is for ordinal (i.e. ranked) data but isn’t yes/no nominal? I was expecting one statistic and p-value for every symptom (7 tests) but the authors presented only one p-value (.897). If all 7 were non-significant, it would be ok to just quickly state that in the Fig caption or associated text. Same for Fig 4/6/7. For Fig 5, by eyeballing anyway, suspect a chi-square on rifampicin would be significant. Basically, am not convinced that the data meets the assumptions for a Mann-Whitney U test to be completed.
Minor points
Line 28: There are some very minor language issues in the abstract that could be further considered. The term “significant” is usually reserved for “statistically significant”. Consider “serious” or another term.
L 34: “high” attitudes might be confusing, even if this drug is not psychoactive. Perhaps “positive”, “favorable” or another word
L 35: Results should be written in the past tense “Pharmacists were”, L105 too
L 38: Results should consistently be reported to tenths place (43.0?%)
L 41: Avoid contractions “it’s” for more formal writing like this, L 191, 194, 244 too.
L 49: Although the term “hard” is used for recreational drugs (e.g. heroin), am not used to seeing “light”. Consider rephrasing.
The intro generally is good and provides the key foundation. Can you include the cost of CBD (prescription and non-prescription) in Croatia? Can you include the drug interactions with a citation?
L 105: The aim of this study was
L 135: The table and figures should be free-standing so that the audience could read just these and have a reasonable understanding of what was done. Can you add additional information to the captions?
L 135: “consummated”? Do you mean consumed?
L 136: Regarding the medications that have interactions with CBD, consider listing the generic names in lower-case.
L 138-41: This content doesn’t fit very well in the statistical analysis section and should go earlier
L 142: This reviewer was unfamiliar with the Survey Monkey sample size calculator. Is this based on a power-analysis or some other factor(s)? What is the algorithm used to determine the required sample size? Could you provide a bit more information?
L 143: “margin of error of 5%”. Do you mean that your p-value was .05?
L 156: Our study consisted
L 167: Am more used to seeing commas for 16,089. Consider double-checking a recent manuscript in this journal for examples
L 179: The “in total n=874.” Can be deleted. Consider including information about completing a cannabidiol survey and the time-frame.
The quality of the figures is generally ok/good although the resolution was only so-so. Is it possible to export with a higher DPI? Fig 1 should have a label (%) on the Y-axis. Reporting the R2 to three decimal places would be sufficient.
This is just a light suggestion to consider but Fig 2 contains a lot, borderline too much, information. Have you thought about calculating a mean/SEM for each group and then running statistics? Alternatively, it might be easier on the audience to have % high (high + very high) reported for each group and then exploratory statistics (chi-sq) testing for between group differences.
Since Fig 1B, C includes the regression, the corresponding text could use a bit more description of this finding.
L 227: In Figure 3 (delete “By). Periods should be reserved for the end of the sentence.
L232: This is a light (stylistic) suggestion. Are the symptoms on the Y-axis arranged in the sequence they appeared on the survey? Consider if arranging them highest % to lowest % (i.e. “waterfall” style). Same for Fig 4.
L 312: Over 1,000 trials? (consider revising)
L 376: “present nearest” is clunky. Consider “current and future health care”
References could use some TLC. Consider double-checking a recent published paper in this journal (non-proper nouns in the article title should be consistently in upper-case (1, 2, 4, 5, 6, 9, 10, 13, 14, etc.).
Comments on the Quality of English LanguageSome minor language issues are noted.
Author Response
The authors wish to thank reviewer for valuable critical input, and hope that the changes introduced in the paper will now meet their acceptance criteria.
Below we described the changes. The changes are highlighted in red. Through track changes it is visible in the text.
Reviewer 2 Report
Comments and Suggestions for Authors
Thank you for the opportunity to review this manuscript. Overall, the work merits publication with some edits as described below. These edits will alleviate some confusion that I believe may be related to translation. Comments 1, 2 and 3 need to be addressed prior to publication. Comment 4 is discretionary.
1. In the Material and Methods section (line 123 and 124) it is stated that attitudes and knowledge questions were assessed on a 5-point Likert scale (from strongly disagree to strongly agree). In Table 2 and 3 the footnotes use a different anchor of 1 – very low; 2 – low; 3 – moderate; 4 – high; 5 – very high. This creates confusion about the actual interpretation in the results section. This is particularly important in line 207 of the results where the authors refer to a “neutral” position on the part of pharmacists. This would only be possible with a strongly disagree to strongly agree scale.
2. In Table 3 the second variable is listed as “CBD treatment is efficient.” Should this actually be “CBD treatment is efficacious.” On line 303 and 304, the authors state that “…believe that CBD treatment is generally effective.” I don’t see that question in the table unless there is an error in wording.
3. Lines 207 to 216 are not clear relative to the statistical significance reported. It is assumed that the p-value is measuring the difference between groups. There is confusing language with a mix of the words “among” and “between”. Stating that there is a significance difference “among” students doesn’t allow the reader to understand where the difference exists. Reference is made to Tables and Figures, but those p-values don’t match up. I think you may mean to use “between” where you use “among”.
4. I am not sure it is valid to conclude that the results confirm CBD consumption being associated with age. I did not see an age demographic collected. This appears to be a conclusion based on the higher usage among students as compared to professionals. Since age was not collected and an association was not reported based on that there could be additional variables in play. Health professionals may be under additional scrutiny or potential drug testing that would mitigate usage as a professional. It may be better to say that usage was higher among students, which represent a younger age group as compared to professionals. Other published work supports…
Comments on the Quality of English LanguageI generally do not edit for grammar. There are some issue that might be related to translation that are included in the comments to the authors.
Author Response

(The authors gave the same response as above.)

Round 2
Reviewer 1 Report
Comments and Suggestions for Authors
My prior concerns (statistics and figures) have been sufficiently addressed.